# In-Hospital Mortality Risk of Transcatheter Arterial Embolization for Patients with Severe Blunt Trauma: A Nationwide Observational Study

**DOI:** 10.3390/jcm9113485

**Published:** 2020-10-28

**Authors:** Masayasu Gakumazawa, Chiaki Toida, Takashi Muguruma, Mafumi Shinohara, Takeru Abe, Ichiro Takeuchi

**Affiliations:** Department of Emergency Medicine, Graduate School of Medicine, Yokohama City University, 4-57 Urafunecho, Minami-ku, Yokohama 232-0024, Japan; msys.gkmzw@gmail.com (M.G.); mgrmtks@gmail.com (T.M.); shinoharamafumi@yahoo.co.jp (M.S.); abet@yokohama-cu.ac.jp (T.A.); itake@myad.jp (I.T.)

**Keywords:** blunt trauma, transcatheter arterial embolization, in-hospital mortality, inter-hospital transfer, age-related differences, haemodynamic stabilization

## Abstract

This study investigated the risk factors for in-hospital mortality of severe blunt trauma patients who underwent transcatheter arterial embolization (TAE). We analysed data from the Japan Trauma Data Bank from 2009 to 2018. Patients with severe blunt trauma and an Injury Severity Score (ISS) ≥ 16 who underwent TAE were enrolled. The primary analysis evaluated patient characteristics and outcomes, and variables with significant differences were included in the secondary multivariate logistic regression analysis. In total, 5800 patients (6.4%) with ISS ≥ 16 underwent TAE. There were significant differences in the proportion of male patients, transportation method, injury mechanism, injury region, Revised Trauma Score, survival probability values, and those who underwent urgent blood transfusion and additional urgent surgery. In multivariable regression analyses, higher age, urgent blood transfusion, and initial urgent surgery were significantly associated with higher in-hospital mortality risk [*p* < 0.001, odds ratio (OR), 95% confidence interval (CI): 1.01 (1.00–1.01); *p* < 0.001, 3.50 (2.55–4.79); and *p* = 0.001, 1.36 (1.13–1.63), respectively]. Inter-hospital transfer was significantly associated with lower in-hospital mortality risk (*p* < 0.001, OR = 0.56, 95% CI = 0.44–0.71). Treatment protocols for urgent intervention before and after TAE and a safe, rapid inter-hospital transport system are needed to improve mortality risks for severe blunt trauma patients.

## 1. Introduction

The mortality and morbidity rates for severe blunt trauma with active bleeding remains significantly high [1,2,3,4]. Early haemodynamic stabilization is an important factor for improving mortality and morbidity for patients with major trauma who are in a haemodynamically unstable state [1,2,5,6,7,8]. Operative management, including damage control surgery, is the first-line treatment for trauma patients with haemodynamic instability [5,6]. Nonetheless, transcatheter arterial embolization (TAE), a non-operative management strategy, is the standard of care for haemodynamically stable trauma patients [7,8,9]. With the advances in endovascular techniques for these patients, some reports have suggested that TAE in conjunction with surgery and/or urgent blood transfusion can more efficiently improve the mortality risk [10,11,12,13]. However, it remains unclear which of the therapeutic strategies among these two interventions is more appropriate for patients with blunt trauma in a haemodynamically unstable state.

Previous studies have reported that age-related physiological and anatomical changes result in differences in the mechanism and outcome of injury in paediatric and geriatric patients with trauma [14,15]. Especially, geriatric patients with trauma have higher mortality and morbidity rates [14,15]. Worldwide, despite the rapidly growing number of trauma patients undergoing TAE, data on the age-related differences and mortality risk associated with TAE for these patients are scarce. We hypothesised that additional urgent interventions for haemodynamic stabilization that are necessitated by ageing might be risk factors for mortality in patients with blunt trauma who undergo TAE.

Therefore, this study aimed to evaluate the risk factors for in-hospital mortality of severe blunt trauma patients who underwent TAE by using a nationwide trauma registry over a recent ten-year study period, while taking the patient’s age into consideration.

## 2. Materials and Methods

### 2.1. Study Setting and Population

This retrospective observational study was conducted based on data obtained from the Japan Trauma Data Bank (JTDB), which registers the data of patients with trauma and/or burns and records the pre-hospitalization and hospital-related information. Data in the JTDB include demographics, comorbidities, injury types, mechanism of injury, means of transportation, vital signs, Abbreviated Injury Scale (AIS) score, prehospital/in-hospital procedures, trauma diagnosis as indicated by the AIS, and clinical outcome. In most cases, physicians who are trained in AIS coding undertake the online registration of individual patient data. The Japan Association for the Surgery of Trauma permits open access and updating of the existing medical information, and the Japan Association for Acute Medicine evaluates the submitted data [16].

In this study, we used a JTDB dataset that included information from 1 January 2009 to 31 December 2018, which initially yielded data for 313,643 patients. The inclusion criteria for this study were the presence of blunt trauma with an Injury Severity Score (ISS) ≥ 16 and patients who underwent TAE. Patients with burns, those with penetrating trauma, those who were dead on hospital arrival, or those with missing key data were excluded from this study. Figure 1 presents a flow diagram of the patient disposition in this study.

### 2.2. Data Collection

We collected the following information from the JTDB: demographic data (age (years), sex, transportation method, and mechanism of injury)(, clinical parameters (AIS score, injury region with AIS ≥ 3, ISS, Revised Trauma Score (RTS) which comprises age, Glasgow Coma Scale score, systolic blood pressure, survival probability, standardized mortality ratio (SMR), usage of computed tomography, blood transfusion within 24 h after admission and initial urgent surgery, and hospitalization location), and outcomes (in-hospital mortality). The survival probability was calculated using the Trauma and Injury Severity Score (TRISS). The SMR was calculated by dividing the in-hospital mortality rate by the mean predicted mortality rate (1—mean survival probability). The outcome measures were in-hospital mortality and risk factors of in-hospital mortality.

### 2.3. Statistical Analysis

This study evaluated: (1) patient characteristics and outcomes during the ten-year study period in ten age groups: and (2) the risk factors associated with in-hospital mortality for patients with blunt trauma who underwent TAE. In the primary analysis conducted to identify characteristics and outcomes of patients who underwent TAE, a Mann–Whitney U test and Kruskal–Wallis test were used to analyze continuous variables, and a chi-square test was used to analyse categorical variables. The age-group-stratified multiple comparisons underwent analysis of variance with Bonferroni corrections.

In the secondary analysis, the following variables that had statistically significantly differences in the primary analysis were included in the multivariate logistic regression analysis. We assessed multicollinearity using the variance inflation factor (VIF) for each covariate. We consider a VIF > 4 to indicate potential multicollinearity. The dependent variable in the multivariate logistic regression was in-hospital mortality. The results of these comparisons are expressed as median and interquartile range (IQR; 25th–75th percentile) for continuous variables and as mean and percentages for categorical variables. All statistical analyses were performed using STATA/SE software, version 16.0 (StataCorp, College Station, TX, USA). A two-tailed *p*-value < 0.05 indicated statistical significance.

### 2.4. Ethics Statement

This study was approved by the institutional ethics committees of Yokohama City University Medical Centre (approval no. B170900003). The approving authority for data access was the Japanese Association for the Surgery of Trauma (Trauma Registry Committee). The requirement of informed consent from the patients was waived owing to the observational nature of the study design.

## 3. Results

During the 10-year study period, 5800 patients (6.4%) from among the total number of blunt trauma cases with an ISS ≥ 16 underwent TAE (Figure 1). These patients were categorized into the following age groups: 0–5 years, 6–15 years, 16–25 years, 26–35 years, 36–45 years, 46–55 years, 56–65 years, 66–75 years, 76–85 years, and ≥ 86 years. There was no significant difference between the age-stratified subgroups in the rate of incidence of TAE in severe blunt trauma patients (1.1%, 5.7%, 8.4%, 8.2%, 7.2%, 6.1%, 5.4%, 5.7%, 6.6%, and 6.9%, respectively, *p* = 0.437), although patients aged 0–5 years had a lower late of incidence of TAE than those in the other subgroups. The study cohort included male patients (*n* = 3585, 62%), patients who were transported from another hospital (*n* = 1053, 18%), and patients with polytrauma (*n* = 4265, 74%). The overall median survival probability, actual in-hospital mortality, and SMR were 70.6%, 17.5%, and 0.60, respectively.

Table 1 shows the patient demographic and outcome data by age group. There were significant differences in the proportion of male patients (*p* < 0.001), transportation method (*p* < 0.001), injury mechanism (*p* < 0.001), injury region with AIS ≥ 3, RTS score (*p* < 0.001), and survival probability (*p* < 0.001) by age group. The older patients had higher in-hospital mortality than the other age groups (*p* < 0.001). Moreover, all patients in the 0–5 years age group survived.

Table 2 shows that there are significant differences in the proportion of patients who underwent urgent blood transfusion (*p* < 0.001) and additional urgent surgery (*p* = 0.001) in the age-stratified subgroups. Older patients had a greater tendency to require blood transfusion.

Table 3 shows the results of multivariate logistic regression analyses. We found multicollinearity between the proportion of patients who were transferred from the scene and from another hospital and between the RTS and survival probability values, and thus we excluded these variables from our multiple logistic regression. Higher age, urgent blood transfusion, and initial urgent surgery were significantly associated with higher odds of in-hospital mortality (*p* < 0.001, odds ratio (OR) = 1.01, 95% confidence interval (CI) = 1.00–1.01; *p* < 0.001, OR = 3.50, 95% CI = 2.55–4.79; and *p* < 0.001, OR = 1.36, 95% CI = 1.13–1.63, respectively). Moreover, the higher survival probability that was calculated on the basis of the TRISS was significantly associated with lower odds of in-hospital mortality (*p* < 0.001, OR = 0.97, 95% CI = 0.96–0.97). Transportation from another hospital was significantly associated with lower odds of in-hospital mortality (*p* < 0.001, OR = 0.56, 95% CI = 0.44–0.71). With regard to the injury region and an AIS score ≥ 3, both chest injury and spinal injury were associated with lower odds of in-hospital mortality (*p* < 0.001, OR = 0.66, 95% CI = 0.54–0.81; *p* = 0.007, OR = 0.71, 95% CI = 0.56–0.91, respectively). Neither sex nor injury mechanism/ISS scores were significantly associated with in-hospital mortality risk.

## 4. Discussion

This is a large nationwide cohort study to identify the risk factors for in-hospital mortality in patients with severe blunt trauma who underwent TAE, and to provide practice-based evidence. This analysis of Japan’s nationwide trauma registry for a ten-year study period showed that patients with higher age, higher injury severity, and urgent intervention, including urgent blood transfusion or surgery, showed an increased risk for in-hospital mortality. In contrast, inter-hospital transfer was significantly associated with survival benefit for patients with severe blunt trauma who underwent TAE.

To the best of our knowledge, there is no national study which has evaluated the characteristics and mortality of patients with severe blunt trauma who underwent TAE in the study cohort, including children and adult, by age-group. In this study, there was no statistically significant difference in the incidences of TAE in patients with severe blunt trauma (Figure 1). Moreover, the SMR was less than 1.00 in all age groups. Therefore, the results of this study suggested that TAE might be an effective therapeutic strategy for severe blunt trauma patients with haemorrhage regardless of age. However, the incidences of TAE in patients with severe blunt trauma who were younger than five years of age were lower than those among the other age groups (1.1% versus 5.4%–8.2%). This finding of a lower incidence of blunt trauma in paediatric patients is in agreement with the findings of earlier studies, which reported that the incidence of TAE for paediatric patients with blunt abdominal or pelvic trauma ranged from 1.4% to 2.1% [17,18,19,20] and was lower than that for adult patients [19]. Despite the difficulties in interpreting the results of this study, the abovementioned finding may be attributed to the higher proportion of patients with head trauma without indications for TAE in the subgroup of younger paediatric severe trauma patients [15].

Previous studies reported that elderly patients with severe trauma had higher mortality and morbidity risks than younger patients [14,15,21]. However, to our knowledge, there was no nationwide cohort study regarding the association between mortality risk and the age of patients with severe blunt trauma who underwent TAE [14]. In this multivariate regression analysis, older age in itself was identified as an independent risk factor for in-hospital mortality risk of TAE. Therefore, in addition to age-related changes in trauma severity, it seems possible that the presence of diminishing activities of daily living, physiological reserves, and pre-existing chronic disease associated with ageing may also affect a higher mortality risk [14,22,23]. In contrast, this study showed that younger patients with blunt trauma had a low risk of in-hospital mortality associated with TAE, and all paediatric patients younger than five years who underwent TAE survived. Moreover, paediatric patients who were younger than 15 years who underwent TAE had a lower SMR than the other age groups; nevertheless, there were no statistically significant differences in the SMR. This result suggests that TAE should be actively considered as a therapeutic option for haemodynamic stabilization in the paediatric population, given the previously reported efficacy and safety of TAE in paediatric patients with abdominal and pelvic trauma [18]. Furthermore, multivariate logistic regression analysis showed higher odds of in-hospital mortality among those with higher injury severity and needing urgent interventions, such as blood transfusion and surgery. Previous studies have showed that the clinical failure and mortality of TAE were associated with the amount of blood loss and the delay between admission and haemodynamic stabilization [10,11]. Although a recent study reported that TAE can be successful in trauma patients with higher injury severity and haemodynamic instability, the results of this study suggest that TAE should be considered as an effective option in the therapeutic strategy for achieving haemodynamic stabilization [10,11,12,13]. Therefore, we should undertake alternative haemodynamic control procedures, such as blood transfusion and/or operative intervention, before and after TAE, to improve the mortality risk for patients with blunt trauma of higher severity and in a haemodynamically unstable state. As mentioned above, the target organ and artery of embolization, performance criteria of TAE, and physical status before and after the TAE procedure were important factors when evaluating the outcome of severe blunt trauma patients who underwent TAE. However, because this additional information was not entered into the JTDB registry, we were unable to perform additional analyses to evaluate whether these factors were associated with in-hospital mortality risk after TAE. In the next research step, analyses with all known risk factors of mortality, including target organ and artery of embolization, performance criteria of TAE, and physical status before and after the TAE procedure, are needed to evaluate the quality of injury care and outcome.

Finally, our results suggested that patients with blunt trauma who underwent TAE would derive a mortality risk benefit from inter-hospital transfer. Previous studies have reported that centralization to tertiary hospitals, which provide higher level care and the availability of medical resources, might be effective for improving the mortality risk in severely injured patients who can avail inter-hospital transfer [24]. The advantages of inter-hospital transfer in blunt trauma who need interventions for haemodynamic stabilization might be attributed to the provision of high-quality care that is associated with TAE by a trained trauma team, including trauma care physicians, interventional radiologists, and surgeons [7]. In contrast, the disadvantages of inter-hospital transfer in severely injured patients were reported and include adverse events during transportation as well as the delays in the decision-making process and provision of definitive care [24,25]. Therefore, these data suggest that we should prepare a timely and safe transportation system, including intra-transport and hospital triage criteria, and a transport team constituted by specialists in haemodynamic stabilization procedure, such that trauma patients benefit maximally and without adverse effects.

This study had several limitations. First, while the study was conducted based on data obtained from a nationwide database with a large sample size, the retrospective study design and the fact that the JTDB had some missing data impaired the precision of the analyses. Second, the proportions of participants and participating hospitals differed across the ten-year study period. Moreover, it is possible that the protocols used in Japanese emergency medical centres evolved during this same ten-year period with the use of damage-control strategies, transfusion protocol, etc. Finally, we could not conduct subgroup analyses based on the location of the accident scene, type of hospital, time interval from injury to initiation of TAE, complications associated with TAE, and the quality of trauma care by hospitals. Several reports have shown that optimal management of trauma patients, according to the abovementioned factors, affects mortality risk. Therefore, future nationwide studies with subgroup analyses should be conducted to establish appropriate management strategies and improvements in the outcomes of patients with blunt trauma and active bleeding. To the best of our knowledge, this study is the first to evaluate in-hospital mortality risks of TAE for blunt trauma in Japan—wherein few systems of injury surveillance and centralization systems for trauma centres have aimed at improving the mortality values of patients with blunt trauma who are in a haemodynamically unstable state.

## 5. Conclusions

Patients with higher age, higher injury severity, and urgent intervention needs, including urgent blood transfusion or surgery, had an increased risk of in-hospital mortality. In contrast, inter-hospital transfer was significantly associated with survival benefit. The results suggest a need to prepare protocols for alternative haemodynamic control procedures, such as blood transfusion and/or operative intervention, before and after TAE and an inter-hospital transport system that can safely and rapidly contribute to improved mortality rates for patients with blunt trauma and higher severity injuries who are in a haemodynamically unstable state.

## Figures and Tables

**Figure 1 jcm-09-03485-f001:**
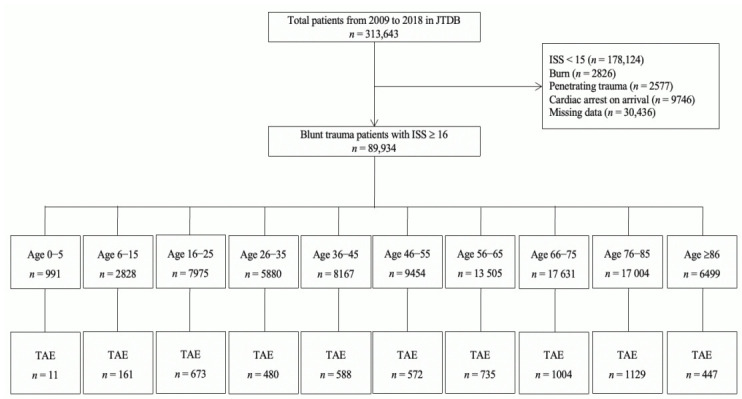
Flow diagram of the study patient disposition. JTDB—Japanese Trauma Data Bank; ISS—Injury Severity Score; TAE—transcatheter arterial embolization.

**Table 1 jcm-09-03485-t001:** Demographics and outcomes of trauma patients who underwent transcatheter arterial embolization and differences between the age-stratified subgroups.

Variables	Total *n* = 5800	Age 0–5 *n* = 11	Age 6–15 *n* = 161	Age 16–25 *n* = 673	Age 26–35 *n* = 480	Age 36–45 *n* = 588	Age 46–55 *n* = 572	Age 56–65 *n* = 735	Age 66–75 *n* = 1004	Age 76–85 *n* = 1129	Age ≥ 86 *n* = 447	*p*-Value
Male, *n* (%)	3585 (62)	4 (36)	101 (63) *	491 (73) *	320 (67) *	410 (70) *	399 (70) *	499 (68) *	598 (60) *	565 (50) *	198 (44) *	<0.001
Transportation, *n* (%)	–	–	–	–	–	–	–	–	–	–	–	–
Transportation from the Scene	4583 (79)	7 (64)	121 (75)	566 (84) *	414 (86) *	501 (85) *	469 (82) *	593 (81) *	759 (76)	844 (75)	309 (69) *	<0.001
Transportation from another Hospital	1053 (18)	3 (27)	31 (19)	92 (14) *	51 (11) *	72 (12) *	85 (15) *	118 (16) *	220 (22)	256 (23)	125 (28) *	<0.001
Injury Mechanism of Blunt Trauma, *n* (%)	–	–	–	–	–	–	–	–	–	–	–	–
Traffic Accident	3187 (55)	9 (81)	91 (57)	427 (63) *	234 (49)	270 (46)	286 (50)	358 (49)	567 (56)	709 (63) *	236 (53) *	<0.001
Fall	1797 (31)	2 (18)	45 (28)	175 (26)	188 (39) *	234 (40) *	192 (34)	246 (33)	318 (32)	281 (25)	116 (26) *	<0.001
Tumble	240 (4)	0	3 (2) *	3 (0.5) *	6 (1) *	7 (1) *	14 (3) *	24 (3) *	38 (4) *	76 (7) *	69 (15) *	<0.001
Injury Region, *n* (%)	–	–	–	–	–	–	–	–	–	–	–	–
Polytrauma	4265 (74)	9 (82)	112 (70)	495 (74)	362 (75)	455 (77)	415 (73)	543 (74)	739 (74)	825 (73)	310 (69)	0.242
Head Injury with AIS ≥ 3	2028 (35)	5 (45)	52 (32)	183 (27) *	127 (26) *	148 (25) *	151 (26) *	248 (34)	417 (42)	517 (46)	180 (40) *	<0.001
Facial Injury with AIS ≥ 3	115 (2)	0	6 (4)	20 (3)	14 (3)	14 (2)	12 (2)	13 (2)	20 (2)	11 (1)	5 (1)	0.055
Neck Injury with AIS ≥ 3	40 (0.7)	0	0	4 (0.6)	2 (0.4)	6 (1)	3 (0.5)	3 (0.4)	8 (0.8)	10 (0.9)	4 (0.9)	0.843
Chest Injury with AIS ≥ 3	3370 (58)	7 (64)	85 (53)	425 (63) *	305 (64) *	374 (64) *	360 (63) *	458 (62) *	567 (56) *	587 (52)	202 (45) *	<0.001
Abdominal and Pelvic Injury with AIS ≥ 3	2394 (41)	10 (91)*	110 (68) *	405 (60) *	247 (51) *	292 (50) *	260 (45) *	316 (43) *	328 (33)	318 (28)	108 (24) *	<0.001
Spinal Injury with AIS ≥ 3	701 (12)	0	10 (6)	73 (11)	63 (13)	95 (16)	65 (11)	90 (12)	121 (12)	136 (12)	48 (11)	0.030
Upper Extremity Injury with AIS ≥ 3	375 (6)	0	10 (6)	49 (7)	37 (8)	59 (10) *	36 (6)	39 (5)	54 (5)	69 (6)	22 (5) *	0.014
Lower Extremity Injury with AIS ≥ 3	3440 (59)	1 (9) *	67 (42) *	307 (46) *	275 (57) *	346 (59) *	333 (58) *	413 (56) *	609 (61) *	749 (66) *	340 (76) *	<0.001
Injury Severity Score, (Median, IQR)	26 (16–38)	18 (9–29)	18 (10–32)	24 (16–34)	25 (16–38)	29 (18–41)	26 (17–36)	26 (17–36)	27 (17–38)	26 (17–41)	26 (16–34)	0.056
Revised Trauma Score, (Median, IQR)	7.55 (6.08–7.84)	6.61 (5.68–7.84)	7.55 (6.61–7.84)	7.55 (6.37–7.84)	7.32 (5.97–7.84)	7.55 (5.97–7.84)	7.55 (6.38–7.84)	7.55 (6.08–7.84)	7.55 (6.17–7.84)	7.55 (5.97–7.84)	7.55 (6.37–7.84)	0.044
Survival Probability, (Median, IQR)	84.7 (53.1–94.3)	95.5 (40.5–99.4)	96.2 (83.8–98.1) *	95.5 (81.4–99.3) *	93.8 (75.7–97.8) *	92.2 (72.2–97.6) *	94.5 (76.9–97.8) *	72.6 (38.7–88.7)	75.7 (41.0–87.8)	72.5 (34.8–88.7)	77.5 (47.2–88.7) *	<0.001
In-Hospital Mortality, *n* (%)	1013 (17.5)	0	9 (5.6) *	66 (9.8) *	55 (11.5) *	74 (12.6) *	77 (13.5) *	128 (17.4) *	182 (18.1) *	295 (26.1) *	127 (28.4) *	<0.001
Standardised Mortality Ratio	0.60	0.00	0.38	0.56	0.59	0.63	0.73	0.47	0.49	0.67	0.84	0.437

The multiple comparisons between the age-stratified subgroups underwent analysis of variance with Bonferroni corrections: * *p* < 0.05, in comparison with the subgroup of patients aged ≥86 years. AIS—Abbreviated Injury Scale; IQR—interquartile range.

**Table 2 jcm-09-03485-t002:** Examination, treatment, and hospitalization of trauma patients who underwent transcatheter arterial embolization and differences between the age-stratified subgroups.

Variables	Total *n* = 5800	Age 0–5 *n* = 11	Age 6–15 *n* = 161	Age 16–25 *n* = 673	Age 26–35 *n* = 480	Age 36–45 *n* = 588	Age 46–55 *n* = 572	Age 56–65 *n* = 735	Age 66–75 *n* = 1004	Age 76–85 *n* = 1129	Age ≥ 86 *n* = 447	*p*-Value
Examination, (Frequency, %)	–	–	–	–	–	–	–	–	–	–	–	–
Computed Tomography	5612 (97)	11 (100)	154 (97)	653 (97)	467 (97)	566 (96)	560 (98)	706 (96)	978 (97)	1091 (97)	426 (95)	0.347
Treatment, (Frequency, %)	–	–	–	–	–	–	–	–	–	–	–	–
Blood Transfusion within 24 h	4881 (68)	10 (59)	104 (43) *	464 (53) *	411 (66)	447 (65) *	471 (68) *	634 (71)	848 (70)	1058 (78)	434 (79) *	<0.001
Initial Urgent Surgery	1195 (21)	1 (9)	33 (21)	151 (22) *	130 (27) *	152 (26) *	137 (24) *	151 (21)	182 (18)	195 (17)	63 (14) *	<0.001
Craniotomy	83 (1)	0	10 (6) *	7 (1)	8 (2)	6 (1)	4 (0.7)	11 (2)	18 (2)	17 (2)	2 (0.5) *	<0.001
Cauterization	98 (2)	0	3 (2)	8 (1)	7 (1)	7 (1)	9 (2)	12 (2)	20 (2)	25 (2)	7 (2)	0.852
Thoracotomy	127 (2)	0	1 (0.6)	17 (3)	14 (3)	17 (3)	14 (2)	23 (3)	17 (2)	18 (2)	6 (1)	0.176
Celiotomy	304 (5)	1 (10)	8 (5)	47 (7) *	33 (7) *	48 (8) *	45 (8) *	39 (5)	40 (4)	35 (3)	8 (2) *	<0.001
Bone Fixation	581 (10)	0	10 (6)	65 (10)	73 (15) *	83 (14) *	73 (13)	76 (10)	97 (10)	74 (7)	30 (7) *	<0.001
Angiotomy	11 (0.2)	0	0	1 (0.2)	1 (0.2)	2 (0.3)	3 (0.5)	1 (0.1)	2 (0.2)	1 (0.1)	0	0.734
Endoscopic Surgery	6 (0.1)	0	0	1 (0.2)	0	0	3 (0.5)	0	1 (0.1)	1 (0.1)	0	0.188
Hospitalization, (Frequency, %)	–	–	–	–	–	–	–	–	–	–	–	–
Intensive Care Unit Admission	5378 (93)	10 (91)	150 (93)	627 (93)	458 (95)	549 (93)	539 (94)	669 (91)	916 (91)	1042 (92)	418 (94)	0.093

The multiple comparisons between the age-stratified subgroups underwent analysis of variance with Bonferroni corrections: * *p* < 0.05, in comparison with the subgroup of patients aged ≥86 years.

**Table 3 jcm-09-03485-t003:** Multivariate logistic regression analysis of in-hospital mortality among trauma patients who underwent transcatheter arterial embolization.

–	Total, *n* = 5800		
–	OR	(95% CI)	*p*-Value
Male	1.07	(0.91–1.26)	0.434
Age, Year	1.01	(1.00–1.01)	<0.001
Transportation	–	–	–
Transportation from Another Hospital	0.56	(0.44–0.71)	<0.001
Injury Mechanism of Blunt Trauma	–	–	–
Traffic Accident	1.08	(0.80–1.40)	0.624
Fall	1.20	(0.88–1.63)	0.246
Tumble	1.35	(0.80–2.27)	0.257
Injury Region	–	–	–
Polytrauma	1.27	(0.94–1.72)	0.118
Head Injury with AIS ≥ 3	1.13	(0.93–1.38)	0.216
Facial Injury with AIS ≥ 3	1.04	(0.65–1.67)	0.865
Neck Injury with AIS ≥ 3	1.73	(0.72–4.11)	0.214
Chest Injury with AIS ≥ 3	0.66	(0.54–0.81)	<0.001
Abdominal and Pelvic Injury with AIS ≥ 3	0.91	(0.75–1.13)	0.373
Spinal Injury with AIS ≥ 3	0.71	(0.56–0.91)	0.007
Upper Extremity Injury with AIS ≥ 3	0.80	(0.57–1.09)	0.154
Lower Extremity Injury with AIS ≥ 3	0.83	(0.68–1.02)	0.078
Survival Probability Calculated with the TRISS	0.97	(0.96–0.97)	<0.001
Treatment	–	–	–
Blood Transfusion within 24 h	3.50	(2.55–4.79)	<0.001
Initial Urgent Surgery	1.36	(1.13–1.63)	0.001

OR—odds ratio; CI—confidence interval; ISS—Injury Severity Score; TRISS—Trauma and Injury Severity Score.

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
