# Peer review of "In-Hospital Mortality Risk of Transcatheter Arterial Embolization for Patients with Severe Blunt Trauma: A Nationwide Observational Study"

_jcm, 2020, doi:10.3390/jcm9113485_

Round 1

Reviewer 1 Report

interesting paper on the mortality risk of trauma patients who underwent transcatheter  artery embolisationinteresting paper on the mortality risk of trauma patients who underwent transcatheter  artery embolisation

Author Response

Reviewer’s comments (in blue) and Answers (in black)

We thank the reviewers for their insightful comments on our paper. We feel that the comments have helped us to significantly improve the manuscript.

Reviewer: 1

Interesting paper on the mortality risk of trauma patients who underwent transcatheter artery embolisation

Response: Thank you for your review of our paper.

Reviewer 2 Report

The authors submitted an interesting work evaluating age-related differences and risk factors associated with mortality in blunt trauma patients undergoing transcatheter arterial embolization, using data from the Japan Trauma Data Bank from 2009 to 2018.

I have some concerns:

Since there seems to be 2 main objectives: “age-related differences” and “risk factors associated with mortality”, the manuscript can be misunderstood. In example, the title mainly refers to age-related differences whilst in section 2.2 data collection, outcome measures are defined as in-hospital mortality and risk factors for in-hospital mortality. Please, revise and unify.

Despite the severity of injury seems high attending to median ISS (table 1), I wonder why the authors included patients with very low ISS (3-8) (the included patients with ISS ≥3). It is very unlikely that these patients need TAE and this might lead to understimate the real incidence of TAE in moderate to severe blunt trauma.

The authors state that they used Mann-Whitney U test and kruskal-Wallis to analyze continuous variables and Chi-Square test to analyze categorical variables. They are comparing multiple categories. Did they use Bonferroni corrections (Dunn’s Test) for multiple comparisons of individual pairs? If then, please state in tables which categories showed statistical differences.  

The first paragraph of the results and figure 1 are repetitive.

The “protective effect” of transfer from another hospital deserves consideration. Did the authors consider a severity of injury adjustment?.

Since the study is conducted in a 10-year period, it is very likely that protocols used in Japan centers evolved in this period with the use of damage-control strategies. This may have resulted in an increased use of TAE in late years. Please, comment in the limitations section.

Author Response

We thank the reviewers for their insightful comments on our paper. We feel that the comments have helped us to significantly improve the manuscript.

Reviewer’s comments (in blue) and Answers (in black)

Reviewer: 2

We thank you for giving us the opportunity to strengthen our manuscript with your valuable comments and queries. We have worked hard to incorporate your feedback and hope that this revision meets your expectations for ensuring high quality in research publication.

Reviewer comments.

  1. Since there seems to be 2 main objectives: “age-related differences” and “risk factors associated with mortality”, the manuscript can be misunderstood. In example, the title mainly refers to age-related differences whilst in section 2.2 data collection, outcome measures are defined as in-hospital mortality and risk factors for in-hospital mortality. Please, revise and unify.

Response: Thank you for this pertinent suggestion. As you pointed out, we mainly evaluated the risk factors associated with in-hospital mortality of transcatheter arterial embolisation for patients with blunt trauma in this study. We have revised the title and Introduction section of our study as follows:

In-hospital mortality risk of transcatheter arterial embolisation for patients with severe blunt trauma: A nationwide observational study

Therefore, this study aimed to evaluate the risk factors for in-hospital mortality of severe blunt trauma patients who underwent TAE by using a nationwide trauma registry over a recent 10-year study period, while taking the patient’s age into consideration.

  1. Despite the severity of injury seems high attending to median ISS (table 1), I wonder why the authors included patients with very low ISS (3-8) (the included patients with ISS ≥3). It is very unlikely that these patients need TAE and this might lead to understimate the real incidence of TAE in moderate to severe blunt trauma.

Response: Thank you for your valuable comment. We completely agree with you and have therefore edited the Methods and Results. In addition, re-performed the statistical analysis for a subgroup of severe blunt trauma patients with ISS ≥16. As a result, we found that there was no significant age-related difference in the rate of incidence of TAE in severe blunt trauma patients (0–5 years; 1.1%, 6–15 years; 5.7%, 16–25 years; 8.4%, 26–35 years; 8.2%, 36–45 years; 7.2%, 46–55 years; 6.1%, 56–65 years; 5.4%, 66–75 years; 5.7%, 76–85 years; 6.6%, and 86 years or older; 6.9%, p=.437). Therefore, we have modified our discussion according to the new results as follows:

In the Results section

These patients were categorised into the following age groups: 0–5, 6–15, 16–25, 26–35, 36–45, 46–55, 56–65, 66–75, 76–85, and ≥86 years. There was no significant difference between the age-stratified subgroups in the rate of incidence of TAE in severe blunt trauma patients (1.1%, 5.7%, 8.4%, 8.2%, 7.2%, 6.1%, 5.4%, 5.7%, 6.6%, and 6.9%, respectively, p=.437), although patients aged 0–5 years had lower late of incidence of TAE those in the other subgroups.

In the Discussion section

In this study, there was no statistically significant difference in the incidences of TAE in patients with severe blunt trauma (Figure 1). Moreover, the SMR was less than 1.00 in all age groups. Therefore, the results of this study suggested that TAE might be an effective therapeutic strategy for severe blunt trauma patients with haemorrhage regardless of age. However, the incidences of TAE in patients with severe blunt trauma who were younger than 5 years of age were lower than those among the other age groups (1.1% versus 5.4–8.2%). This findings of a lower incidence of blunt trauma in paediatric patients is in agreement with the findings of earlier studies, which reported that the incidence of TAE for paediatric patients with blunt abdominal or pelvic trauma ranged from 1.4% to 2.1% [17-20] and was lower than those for adult patients [19]. Despite the difficulties in interpreting the results of this study, the abovementioned findings may be attributed to the higher proportion of patients with head trauma without indication for TAE in the subgroup of younger paediatric patients [21].

  1. The authors state that they used Mann-Whitney U test and kruskal-Wallis to analyze continuous variables and Chi-Square test to analyze categorical variables. They are comparing multiple categories. Did they use Bonferroni corrections (Dunn’s Test) for multiple comparisons of individual pairs? If then, please state in tables which dcategories showed statistical differences.

Response: Thank you for this pertinent suggestion. Accordingly, we have added the expression on the multiple comparisons in the Methods section and tables as follows:

In the Methods section

The age-group-stratified multiple comparisons were analysed by the analysis of variance with Bonferroni correction.

In the Tables

The multiple comparisons between the age-stratified subgroups were analysed by the analysis of variance with Bonferroni corrections: *p<0.05 in comparison with the subgroup of patients group aged ≥86 years.

  1. The first paragraph of the results and figure 1 are repetitive.

Response: Thank you for your feedback. Accordingly, we have revised the first paragraph of the results and figure 1 as follows:

These patients were categorised into the following age groups: 0–5, 6–15, 16–25, 26–35, 36–45, 46–55, 56–65, 66–75, 76–85, and ≥86 years. There was no significant difference between the age-stratified subgroups in the rate of incidence of TAE in severe blunt trauma patients (1.1%, 5.7%, 8.4%, 8.2%, 7.2%, 6.1%, 5.4%, 5.7%, 6.6%, and 6.9%, respectively, p=.437), although patients aged 0–5 years had a lower late of incidence of TAE than those in the other subgroups.

  1. The “protective effect” of transfer from another hospital deserves consideration. Did the authors consider a severity of injury adjustment?

Response: Thank you for your comment. We concur with your suggestion that a severity of injury may be one of the confounding factors when we analyse the association between in-hospital mortality in patients who underwent TAE and inter-hospital transportation. Therefore, we have carried out a multivariate regression analysis wherein the in-hospital mortality was set as a dependent variable and 18 potential confounder factors, including the interhospital transportation and severity of injury (survival probability calculated by the TRISS method), were included as explanatory variables in order to exclude the influence of the confounding factors. The findings showed that inter-hospital transportation had a protective effect for in-hospital mortality of severe blunt trauma patients who underwent TAE.     

  1. Since the study is conducted in a 10-year period, it is very likely that protocols used in Japan centers evolved in this period with the use of damage-control strategies. This may have resulted in an increased use of TAE in late years. Please, comment in the limitations section.

Response: We appreciate your comment on this point. Accordingly, we have added this point in the limitation section as follows:

Moreover, as the retrospective study investigated a 10-year period, it is possible that the protocols used in Japanese emergency medical centres evolved during this period with the use of damage-control strategies, the massive transfusion protocol, and so on.

Reviewer 3 Report

Dr. Gakumazawa et al are ought to be congratulated for targeting an interesting topic related to the mortality in patients treated with transcatheter arterial embolization after blunt trauma. While the topic is interesting there are several concerns with the paper that would need addressing

The authors made up a monocenter observational study of a national database comparing the in hospital mortality of patients undergoing tae after blunt trauma defined by an ISS of 3 or more.

Why  was the cut off of an ISS of 3 or more chosen? This does not reflect at any point an injury which should need an TAE?

What was performed? Which injuries were treated with TAE? Which vessels were addressed?

What was the cut off in blood loss, blood pressure or shock classification to perform TAE?

Blunt trauma can be anything, and with an ISS of 3 or more there is a big lack of information about the patient itself.

What was the trigger for an Interhospital Transport? What has been done in first hospital?

The Survival probability measured by the TRISS is in every group really high. In contrast to this the mortality is higher than expected in nearly every age

I do not recommend publishing this publication.

Author Response

We thank the reviewers for their insightful comments on our paper. We feel that the comments have helped us to significantly improve the manuscript.

Reviewer: 3

We wish to express our strong appreciation to the reviewers for their insightful comments, which have helped us significantly improve the paper.

Reviewer comments.

  1. The authors made up a monocenter observational study of a national database comparing the in hospital mortality of patients undergoing tae after blunt trauma defined by an ISS of 3 or more. Why was the cut off of an ISS of 3 or more chosen? This does not reflect at any point an injury which should need an TAE?

Response: Thank you for your valuable comment. We strongly agree with you, and have therefore edited the Methods and Results of the additional work suggested by the reviewers. We have performed again statistical analysis for severe blunt trauma patients with ISS ≥16. As the results, we have revised our discussion according to our results.

  1. What was performed? Which injuries were treated with TAE? Which vessels were addressed?
  2. What was the cut off in blood loss, blood pressure or shock classification to perform TAE?
  3. Blunt trauma can be anything, and with an ISS of 3 or more there is a big lack of information about the patient itself.

Response: We agree on the valuable suggestion of showing the target region and artery of embolization, performance criteria of TAE, and physical status before and after TAE procedure in our study. However, because this additional information was not input in Japan Trauma Data Bank (JTDB) registry, we are unable to show this additional information in our study. Therefore, we have modified the reason why we are unable to show this information as the Discussion section in our study as follows:

As mention above, the target organ and artery of embolization, performance criteria of TAE, and physical status before and after TAE procedure are important factors when we evaluate the outcome of severe blunt trauma patients underwent TAE. However, because these additional information were not input in the JTDB registry, we were unable to performed the additional analysis showing whether these factors were associated with the in-hospital mortality after TAE. In the next research step, we should do analyses with all known risk factors of mortality including target organ and artery of embolization, performance criteria of TAE, and physical status before and after TAE procedure to evaluate the quality of injury care and outcome.

  1. What was the trigger for an Interhospital Transport? What has been done in first hospital?

Response: We agree that the additional information on the interhospital transport as you pointed out would be valuable. However, because this information was not input in the JTDB registry we were unable to performed the discussion regarding this point.

  1. The Survival probability measured by the TRISS is in every group really high. In contrast to this the mortality is higher than expected in nearly every age.

Response: Thank you for your valuable comment. We agree with you and have therefore calculated the standardised mortality ratio (SMR). Therefore, we have added the results in the main text and table 1 as follows:   

The overall median survival probability, actual in-hospital mortality, and SMR were 70.6%, 17.5%, and 0.60, respectively.

Round 2

Reviewer 2 Report

The authors have appropriately addressed my queries.

I detected one mistake in the first sentence of the results section. It reads ISS ≥ 14 and it should read ISS ≥ 16 (as described earlier in the methods section).

Author Response

Reviewer: 2

Reviewer comments.

  1. I detected one mistake in the first sentence of the results section. It reads ISS ≥ 14 and it should read ISS ≥ 16 (as described earlier in the methods section).

Response: Thank you for your feedback. Accordingly, we have revised ISS ≥ 14 to ISS ≥ 16 as follows:

During the 10-year study period, 5800 patients (6.4%) from among the total number of blunt trauma cases with an ISS ≥ 16 underwent TAE (Figure 1).

Reviewer 3 Report

The revisions have been adressed by the authors point by point.

There are still data about the TAE missing, but not available in their databank,so this cannot further be answered with the underlying data.

Author Response

Dear Reviewer

We thank you for giving us the opportunity to strengthen our manuscript with your valuable comments and queries.